# Dissociation of tone merger and congenital amusia in Hong Kong Cantonese

**Caicai Zhang**[ID][1]*, **Oi-Yee Ho**[2], **Jing Shao**[3,4], **Jinghua Ou**[5], **Sam-Po Law**[6]

**1** Research Centre for Language, Cognition, and Neuroscience, Department of Chinese and Bilingual Studies, The Hong Kong Polytechnic University, Hong Kong SAR, China, **2** Ear Institute, University College London, London, United Kingdom, **3** Department of English Language and Literature, Hong Kong Baptist University, Hong Kong SAR, China, **4** Shenzhen Institutes of Advanced Technology, Chinese Academy of Sciences, Shenzhen, China, **5** Department of Linguistics, University of Chicago, Chicago, IL, United States of America, **6** Unit of Human Communication, Development, and Information Sciences, The University of Hong Kong, Hong Kong SAR, China

* caicai.zhang@polyu.edu.hk

## Abstract

While the issue of individual variation has been widely studied in second language learning or processing, it is less well understood how perceptual and musical aptitude differences can explain individual variation in native speech processing. In the current study, we make use of tone merger in Hong Kong Cantonese, an ongoing sound change that concerns the merging of tones in perception, production or both in a portion of native speakers, to examine the possible relationship between tone merger and musical and pitch abilities. Although a previous study has reported the occurrence of tone merger independently of musical training, it has not been investigated before whether tone-merging individuals, especially those merging tones in perception, would have inferior musical perception and fine-grained pitch sensitivities, given the close relationship of speech and music. To this end, we tested three groups of tone-merging individuals with various tone perception and production profiles on musical perception and pitch threshold tasks, in comparison to a group of Cantonese speakers with congenital amusia, and another group of controls without tone merger or amusia. Additionally, the amusics were compared with tone-merging individuals on the details of their tone discrimination and production profiles. The results showed a clear dissociation of tone merger and amusia, with the tone-merging individuals exhibiting intact musical and pitch abilities; on the other hand, the amusics demonstrated widespread difficulties in tone discrimination yet intact tone production, in contrast to the highly selective confusion of a specific tone pair in production or discrimination in tone-merging individuals. These findings provide the first evidence that tone merger and amusia are distinct from each other, and further suggest that the cause of tone merger may lie elsewhere rather than being driven by musical or pitch deficits. We also discussed issues arising from the current findings regarding the neural mechanisms of tone merger and amusia.

**Data Availability Statement:** All relevant data are within the paper and its Supporting Information files.

**Funding:** This work was supported by grants from the Research Grants Council of Hong Kong (ECS:

25603916; https://www.ugc.edu.hk/eng/rgc/), the National Natural Science Foundation of China (NSFC: 11504400; http://www.nsfc.gov.cn/), and the Departmental General Research Funds (Cross-institutional collaboration; https://www.polyu.edu.hk/cbs/web/en/) to CZ. The funders had no role in study design, data collection and analysis, decision to publish, or preparation of the manuscript.

**Competing interests:** The authors have declared that no competing interests exist.

## Introduction

Individual variation is a fundamental issue in research on speech processing and learning. Whereas a plethora of studies have probed into individual differences in second language learning [1–6], individual differences in native speech processing are less well understood and traditionally treated as noise to be minimized in the data. Their potential contribution to theoretical accounts of speech processing and learning is not properly recognized until recently [7–10]. A reason for this problem may be related to the implicit assumption that adult speakers are comparably skilled at processing speech sounds in their native language [11,12]. An intriguing case that challenges this assumption and clearly demonstrates individual differences among adult speakers in native speech processing is language variation and change (e.g., tone merger).

### Tone merger in Hong Kong Cantonese

Tone merger is a type of ongoing sound change that concerns the merging of certain pairs of tones in either production or perception in Hong Kong Cantonese [13,14]. There are six contrastive tones on open syllables in Hong Kong Cantonese: T1 –high level tone, T2 –high rising tone, T3 –mid level tone, T4 –low falling/extra low level tone, T5 –low rising tone, and T6 –low level tone. Accumulating evidence has indicated that three pairs of tones, T2-T5, T3-T6 and T4-T6, are in the process of merging by some native speakers of Hong Kong Cantonese, perhaps due to similar acoustic features such as pitch height/contour between tones in each pair [13–15]. Close examination of the merging patterns reveals a complex mismatch between speech perception and production. While some individuals have been found to confuse T2-T5 and T3-T6 in production but not in perception (i.e., [+per][-pro]), other individuals have demonstrated confusion for T4-T6 in perception but not in production (i.e., [-per][+pro]), and yet other individuals have shown difficulties distinguishing T2-T5 in both perception and production (i.e., [-per][-pro]) [13,16–18]. Overall, three types of tone merger were reported: [+per][-pro] indicating better perception but poorer production, [-per][+pro] indicating poorer perception but better production, and [-per][-pro] where both perception and production are poorer [13,16–18]. Merging of other tone pairs has rarely been reported. A recent large-scale investigation of tone perception and production performance in 120 Hong Kong Cantonese speakers ranging from 20 to 58 years old confirmed these merging patterns [14]. Among the 120 participants, 17.5% demonstrated [-per][-pro] of T2-T5 and another 17.5% demonstrated [-per][+pro] of T2-T5; 46.67% of the participants demonstrated [+per][-pro] of T3-T6; 13.33% demonstrated [+per][-pro] of T4-T6 and another 16.67% demonstrated [-per][+pro] of T4-T6. These tonal variations may be deemed as sub-varieties of standard Hong Kong Cantonese [13–15].

Despite the mounting descriptive studies on tone merger in Hong Kong Cantonese, the underlying cause of tone merger and its relationship with individual differences in perceptual and cognitive aptitude is largely unresolved. Mok and Zuo [19] examined the relationship between tone merger and musical training. Cantonese speakers who merge tones were selected based on production confusion (i.e., not producing six Cantonese tones with clear distinctions); ensuing acoustic analysis confirmed reduced distance in F0 offset between T2 and T5 in the merging group compared to the non-merging group, in addition to diminished overall F0 separation among the five Cantonese tones (T1 was excluded for the reason that it does not participate in tone merger). The merging participants were further divided into merging musicians (N = 10) based on self-report of having received more than seven years of formal musical training and had regular musical practice in the last two years, and into merging nonmusicians (N = 11) if a participant had no more than two years of casual musical experience and did not

play music regularly in the last two years. Intriguingly, there was no group difference between merging musicians and nonmusicians in either tone production or the discrimination accuracy and reaction time (RT) of tones based on an AX discrimination task. These results imply that tone merger is likely to be a domain-specific phenomenon, relatively free from the influence of musical training.

While the results of Mok and Zuo [19] are informative, there are several limitations with this study. First, the merging and non-merging participants were selected based on the confusion of tones in *production*, and it is not clear whether they showed any *perceptual confusion* of tones, given that there are different types of merging individuals as discussed above. Although it was suggested that the merging group (irrespective of their musical training) showed some discrimination difficulty compared to the non-merging group, displaying slower RT in the discrimination of tone pairs T2-T5, T3-T6, and T4-T6, these differences would not have survived correction for multiple comparisons. Second, there is a lack of objective measurement of musical aptitude in the merging and non-merging individuals in the previous study, and discrete group labels (musicians and non-musicians) were assigned based on the participants' self-report of past musical training. For this reason, it is unknown whether musical aptitude objectively measured as continuous variables would exhibit a clearer relationship with the participants' tone merging behavior. Thirdly and most importantly, it remains unknown whether merging individuals, especially those who confuse tones perceptually, would show *inferior* musical abilities and *worse* domain-general sensitivities to fine-grained pitch height and contour differences compared to non-merging individuals.

Previous studies have revealed a mixed picture when it comes to the question of whether tonal language speaker with musical training would have an additive advantage in their native speech perception. In contrast to the consistent findings of musical training or higher musical aptitude associated with better non-native tone perception or learning in *non-tonal language speakers* [3–6,20], results on the effect of musical training in *tonal language speakers* are mixed and inconclusive. While some study has revealed an additive effect of musical abilities and tonal language experience, reporting better native tone perception in Cantonese speakers with absolute pitch than those without [21], other studies have found no evidence for such additive effects [22], or that the musician advantage is restricted to more trivial aspects such as the discrimination of within-category pitch distinctions [9,23] or processing speed [8]. It is likely that the strong influence of top-down language knowledge in tonal language speakers has confined the effect of musical aptitude in native tone perception. The top-down influence may also explain the lack of musician advantage in tone-merging individuals in Mok and Zuo [19] to some extent. In contrast to the mixed picture, it remains an open question whether tone-merging individuals, especially those confusing tones perceptually, would have degraded musical abilities and/or domain-general pitch sensitivity. Although top-down language knowledge might offer some protection, the influence of inferior musical abilities and/or domain-general pitch sensitivity is likely to be detectable. Indeed, tone confusion has been reported in Cantonese speakers with a musical disorder (see the next subsection on congenital amusia), which provides justification for this hypothesis. Without examining musical perception and pitch sensitivity in tone-merging individuals, it would not be possible to conclude confidently that tone merger is domain specific with little influence from the musical domain or domain-general pitch processing ability.

There is some evidence that tone merger in Hong Kong Cantonese can be partially explained by non-linguistic factors [12,24]. For instance, Ou and colleagues [12] found that visual working memory (as assessed by Coding and Cancellation of WAIS-IV) and auditory attentional switching ability (as assessed by Elevator Counting with Reversal in the Test of Everyday Attention) significantly explained the overall response latencies of tone

discrimination in non-merging and merging individuals. The authors argued that working memory ability is important for maintaining information for speech discrimination, and auditory attention may play a role in building language-specific phonological representations by focusing the listeners' attention on language-relevant information. Thus, individual differences in non-linguistic cognitive abilities of working memory and attentional control were found to explain the speech processing speed. This result somewhat echoes an earlier finding that individuals' cognitive processing style, as indexed by their scores of the Autism Spectrum Quotient including attention switching and attention to detail, can explain their perceptual compensation for context-induced acoustic variation to retrieve the intended speech sound category [25]. However, note that none of these studies have examined individual differences in musical perception and pitch sensitivity in tone-merging individuals, and whether musical and pitch aptitude can explain their tone merging performance in a domain-general manner.

## Tone confusion in individuals with congenital amusia

Similar confusion of lexical tones has been reported in individuals with a musical disorder, namely congenital amusia (amusia hereafter) [26–31]. Amusia is a neurodevelopmental disorder that affects musical pitch processing without the presence of brain injury in up to 4% of the population [32–36]. Individuals with amusia (hereafter amusics) demonstrate difficulties in discriminating musical pitch and perceiving patterns and directions of pitch change [37,38]. They also exhibit reduced sensitivity toward detecting fine-grained pitch differences in pitch threshold tasks [37,39], and inferior pitch memory [40–42]. Recent studies have consistently reported inferior performance of amusics in perceiving lexical tones [26,28,30,31,35,43–45]. With regard to Hong Kong Cantonese, Liu and colleagues [26] reported overall lower accuracy in discriminating native tone pairs with different magnitudes of pitch differences in Cantonese speakers with amusia than musically intact controls. Notably, the group factor did not interact with the tone pair factor, which seemed to suggest an overall impairment of amusics regardless of the perceptual difficulty of different tone pairs. But there is some evidence that the tone perception impairment in amusics may be more pronounced in tone pairs with smaller pitch differences, e.g., high rising tone /ji25/ (T2) vs. low rising tone /ji23/ (T5), and to a lesser extent mid level tone /ji33/ (T3) vs. low level tone /ji22/ (T6) in Cantonese [46], exactly the tone pairs that have been reported to participate in tone merger. Contrary to their poor performance in tone perception, amusics' speech pitch production is mostly preserved [47–49]. It has been found that Mandarin-speaking amusics could produce native lexical tones accurately [35,50], albeit exhibiting poor performance in imitating intonation with more pitch interval and contour errors [49]. A recent study reported that Cantonese-speaking amusics were able to produce native tones with comparable fundamental frequency (F0) trajectories as controls, showing normal acoustic distinctions between tones [26].

Studies on the influence of amusia on speech processing fall within a broader line of research on the relationship of speech and music. Some researchers have hypothesized that speech and music share processing mechanisms, pointing out that there are close parallels between speech and music in the use of basic acoustic attributes including pitch [51–54]. In contrast, other researchers argue that these two are separate, encapsulated domains [55,56]. A large amount of studies have reported bidirectional cross-domain transfer of speech and music, which provides some support for the first view of shared processing mechanisms between speech and music. For instance, a plethora of studies have demonstrated an advantage of musicianship or musical training in speech processing and learning [20,57–60], whereas amusia is linked to inferior speech processing [26,28,30,31,35,43,44], demonstrating the music-to-speech influence. Meanwhile, long-term experience with a tone language has been

found to benefit musical processing, including the acquisition of absolute pitch, a rare musical ability [61], and the performance on musical melody discrimination [62], exemplifying the speech-to-music influence. In light of the widely attested bidirectional transfer of speech and music, it is thus of particular interest to examine whether inferior abilities of native tone perception and/or production in tonal language speakers (i.e., tone-merging individuals in Hong Kong Cantonese) are associated with worse musical abilities and domain-general pitch sensitivities.

However, the relationship between tone merger and amusia has not been systematically examined before, despite the seemingly similar behavior of tone confusion, at least in perception, between Cantonese speakers who merge tones and those who have amusia. On the one hand, as mentioned above, it is yet unknown whether tone-merging individuals have inferior musical abilities and pitch sensitivities, and if yes, whether individual differences in these domain-general perceptual abilities can explain the tone merging behavior. On the other hand, although amusics have been found to perceptually confuse those pairs of tones that are in the process of merging, it is unclear whether their performance of tone perception (and production) is comparable to that of the merging individuals. While the previous studies have provided circumstantial evidence for the similarity in tone perception performance between amusics and merging individuals, it is necessary to compare the two groups within a single design, employing the same set of tone perception and production tasks. Last, although one study has reported intact tone production in Cantonese amusics [26], this result calls for replication by more studies. Answering these questions on the relationship of tone merger and amusia will not only lead to a better understanding of the underlying cause of tone merger, but also shed some light on the broader issue of the relationship of speech and music.

## The current study

In the current study, we aimed to examine the musical perception abilities and domain-general pitch sensitivities in tone-merging individuals and compared their tone perception and production performance to that of a group of amusics within a single design, in order to better understand the relationship between tone merger and amusia. We recruited three groups of merging individuals with different profiles of tone perception and production confusion (a group with [+per][-pro] of T2-T5, a group with [-per][+pro] of T4-T6, and a group with [-per][-pro] of T2-T5), to examine if there is any difference between individuals who merge tones in *perception* and those merging tones in *production* or in both in their musical abilities and pitch sensitivities. It is likely that inferior musical abilities and pitch sensitivities, if any, would tend to be manifested in tone-merging individuals with poor perception. Note that we selected individuals who confuse T2-T5 because this tone pair exhibited various patterns of confusion in perception and production ([+per][-pro] and [-per][-pro]). However, individuals who confuse T2-T5 in perception but not in production ([-per][+pro]) are rarely found in our pool of participants [24,63] or in previous reports of tone merger [13,16–18]. For this reason, the T4-T6 pair that showed [-per][+pro] was used instead. The T3-T6 pair was not selected because this pair primarily shows confusion in production but not in perception ([+per][-pro]). These three groups of tone-merging individuals ([-per][+pro] (T4-T6); [+per][-pro] (T2-T5); [-per][-pro] (T2-T5)) were tested on a series of musical perception and pitch threshold tasks, in contrast to a group of amusics for comparison, and another group of controls who do not exhibit tone merger or amusia. Additionally, the amusics were compared with the tone-merging individuals with the same set of tone discrimination and production tasks to verify whether the amusics' tone confusion patterns are similar to or deviant from those of the tone-merging individuals. In summary, we aimed to address the following three questions in this

study: (1) whether tone-merging individuals have poor musical perception and domain-general pitch sensitivity, and if yes, whether their performance approaches that of amusics; (2) whether individual differences in musical abilities and domain-general pitch sensitivities in the tone-merging individuals can explain their tone merging behavior in perception or production; and (3) whether the tone confusion patterns of amusics in tone discrimination and production are similar to those of tone-merging individuals.

## Materials and methods

### Ethics statement

The experimental procedures were approved by the Human Subjects Ethics Sub-committee of The Hong Kong Polytechnic University (PolyU) (Application number: HSEARS20160216001). Informed written consent was obtained from the participants in compliance with the experiment protocols.

### Participants

A total of 73 participants were recruited to participate in the current study. Table 1 shows the demographic info of the participants. All participants were right-handed undergraduates in universities in Hong Kong, with Hong Kong Cantonese as their native language. There was no reported hearing impairment, communication disorders and history of brain injury in any participant. They were divided into five groups–[+per][-pro] (T2-T5), [-per][+pro] (T4-T6), [-per][-pro] (T2-T5), amusics, and controls, based on the criteria described below. Note that [-pro] or [-per] does not imply chance-level performance in the tone-merging group. See the criteria described below.

There is no standardized test for identifying tone-merging individuals, and previous studies have selected merging individuals using either a reading aloud task [15,19], or both a reading aloud task and an AX tone discrimination task [12,14,24,64], based on the criteria of production or perceptual confusion of tones. Note that no musical perception task has been included as part of the identification of merging individuals in the previous studies. In the current study, we adopted the same task and criteria as in [12], and used a lexical tone production task and an AX tone discrimination task. These two tasks were also used to assess tone production and discrimination abilities in amusical participants and to compare tone-merging individuals with amusics. Details of the two tasks were described in the Stimuli and Procedure section below.

Out of the 73 participants, eight controls (4M, 4F), seven [+per][-pro] (T2-T5) participants (3M, 4F), four [-per][+pro] (T4-T6) participants (2M, 2F), and six [-per][-pro] (T2-T5) participants (1M, 5F) were recruited from the participant pool of merging and non-merging individuals who had participated in the previous study [12]. The rest of the participants were

**Table 1. Demographic information of the five groups of participants.**

|  | No. | Mean age (SD) |
|---|---|---|
| Amusics | 19 (8M, 11F) | 22.05 (2.30) |
| Controls | 15 (7M, 8F) | 24.60 (4.10) |
| [+per-pro] (T2-T5) | 13 (7M, 6F) | 20.92 (2.14) |
| [-per+pro] (T4-T6) | 11 (5M, 6F) | 22.09 (2.55) |
| [-per-pro] (T2-T5) | 15 (2M, 13F) | 21.07 (1.49) |

M = male, F = female.

recruited and identified at PolyU. For the remaining participants, their tone production accuracy was evaluated by two phoneticians with Hong Kong Cantonese as their native language, focusing on the production of T2, T4, T5 and T6 that participate in tone merging. There was a high inter-rater agreement as indicated by the result of Kappa Cohen's test ( = .893, $p < .001$). The participants' tone discrimination accuracy was measured from the AX discrimination task. For both tone production and discrimination, an 80% cut-off was used. Participants who had an accuracy above 80% in T2-T5 discrimination but below 80% in T2-T5 production were classified into the [+per][-pro] group; those who achieved lower than 80% accuracies in both T2-T5 perception *and* T2-T5 production were assigned into the [-per][-pro] group; the [-per][+pro] group included participants who failed to score over 80% in T4-T6 perception but with an accuracy of over 80% in T4-T6 production; finally, controls were participants who did not exhibit any kind of tone merger or amusia (see below). A somewhat high cut-off (80%) was used in the current study in order to capture more variances within each tone-merging group. As the tone merger is still on going and incomplete in the community, there is a broad spectrum of individual variations in the accuracy of tone production and perception. Indeed, the majority of individuals fall in between the two ends, i.e., 100% accuracy and chance-level performance [12,24,63]. Thus, the 80% cutoff allowed us to capture a broader and also more faithful representation of individual variations in tone-merging individuals. Importantly, these merging groups did exhibit the aforementioned tone confusion according to the results of the tone production and discrimination tasks (see the Results section below).

For the determination of amusia, the Montreal Battery of Evaluation of Amusia (MBEA), a standardized test to identify amusics [65,66] was used, where participants obtaining a global score of 71.7 or less [35] were regarded as amusic individuals. Details of the MBEA are explained in the Stimuli and Procedure section below. Note that tone perception or production confusion is not part of the diagnostic criteria of amusia.

## Stimuli and procedure

**MBEA.** The MBEA consists of six subtests [65]. The first three subtests are pitch-related (scale, contour and interval), followed by two subtests that are rhythm-related (rhythm and metre), and a last subtest that assesses incidental melodic memory (memory). During the first four subtests, participants were required to indicate whether two presented melodies are same or different. For the Metre subtest, participants had to indicate if the melody played is a march or waltz. Lastly, during the Memory subtest, participants had to decide whether they have heard the presented melodies or not in the previous five subtests. All stimuli were presented on E-Prime 2.0 in a soundproof room through JVC HA-D610 stereo headphones binaurally at a comfortable listening level selected by the participant at the beginning and kept constant throughout the rest of the experiment. All responses were made by pressing the left (same/march/have heard before) or right (different/waltz/have not heard before) arrow on a computer keyboard in accordance with the instructions.

**Pitch threshold.** The pitch threshold task was identical to that reported in [67] and followed the design of [39]. The purpose of this task is to assess a participant's sensitivity to just-noticeable differences in pitch height and contour embedded in speech and nonspeech stimuli. We employed a *stimulus type* (speech and nonspeech) × *contour type* (discrete and gliding) design. For the speech condition, the syllable [ji] recorded by a male native speaker of Hong Kong Cantonese was used, and various pitch manipulations (see below) were imposed on the base syllable. For the nonspeech condition, complex tones, which carried the same F0 cues as the speech condition, were generated using PRAAT [68]. A 15-ms amplitude ramp was applied to the onset and offset of the complex tones to adjust for rise or decay time. All stimuli were

250 ms in duration. The discrete condition included a pair of sounds with flat pitches that differ in pitch height, whereas the gliding condition included a pair of sounds with falling and rising pitch or rising and falling pitch.

For the discrete condition, the stimuli were comprised of a standard sound with a flat pitch at 100 Hz (which is close to the original pitch produced by the male speaker) and 82 target sounds also with flat pitches ranging from 100.07 Hz to 178.17 Hz (0.01 to 10 semitones). The standard sound was paired with one of the 82 target sounds with a 250 ms inter-stimulus interval, where the standard could occur either before or after the target. The largest and smallest difference between the standard and target was 10 and 0.01 semitones respectively. For the gliding condition, the stimuli consisted of 82 rising and 82 falling glides with pitch excursion sizes between 0.01 and 10 semitones. Centred on 100 Hz, the largest rising glide started at 78.67 Hz and ended at 133.12 Hz, while the smallest rising glide started at 100.09 Hz and ended at 100.58 Hz. The falling glides had the same F0 excursion sizes as the rising ones but in the opposite direction. A rising and falling sound with the same F0 excursion size was paired with a 250 ms inter-stimulus interval, where the rising sound could occur either before or after the falling sound. Fig 1 provides an illustration of the F0 manipulations in the discrete and gliding conditions.

Stimuli in the four conditions were presented in four separate blocks that were counterbalanced in order. The task was two-alternative forced choice, where the participants were instructed to indicate the pitch pattern of the stimulus pair (High-Low or Low-High for discrete pairs, and Rising-Falling or Falling-Rising for gliding pairs) by clicking the left (High-Low/Rising-Falling) or right (Low-High/Falling-Rising) arrow buttons on the computer keyboard. Experimental trials were presented with an adaptive tracking procedure. The experiment began with trials with a 10-semitone difference (10 semitones in F0 height for the discrete condition or in F0 excursion size for the gliding condition) and followed a 'two-down one-up' staircase adaptive tracking method. The semitone difference was initially reduced by 1 semitone upon two consecutive correctly judged trials, and the reduction size was adjusted to 0.1 semitone when the semitone difference reached 1 semitone, and further adjusted to 0.01 semitone when the semitone difference reached 0.1 semitone. An incorrect trial led to a reversal to the previous semitone difference. The experiment ended after 14 reversals, and a participant's pitch threshold was calculated from the last six reversals as the mean pitch difference in semitones between the standard and target stimuli in the case of discrete pitches, and the mean pitch excursion size of the gliding pitches, separately for the speech and nonspeech condition.

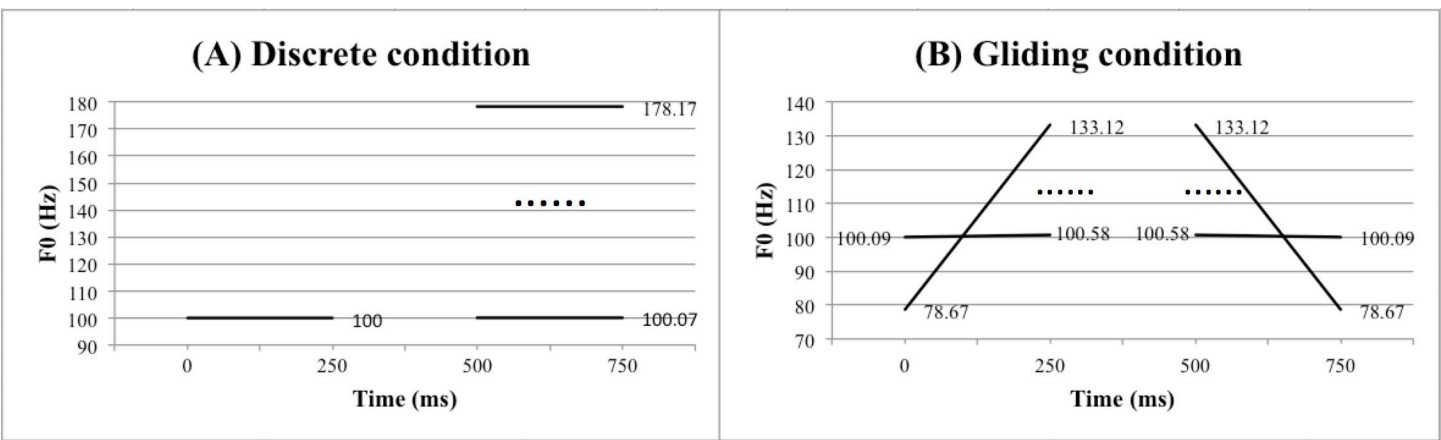

**Fig 1.** An illustration of the F0 manipulations in the discrete (A) and gliding (B) condition in the pitch threshold task. See text for a detailed description of this task.

E-Prime 1.1 was used to control the presentation of stimuli and recording of responses. All stimuli were presented in a soundproof room through JVC HA-D610 stereo headphones binaurally at a comfortable listening level selected by the participant at the beginning and kept constant throughout the experiment.

**Tone production.**   The tone production task was identical to that described in [12]. It was administered before the tone discrimination task to prevent any priming effect on tone production, since participants might be influenced by the auditory characteristics of tone stimuli in the discrimination task and carry them over into production.

The stimuli were the syllable [fu] carrying the six tones, which yielded six meaningful words in Cantonese (夫 [fuː˥] 'husband', 苦 [fuː˧˥] 'bitter', 褲 [fuː˧] 'trousers', 符 [fuː˨˩] 'symbols', 婦 [fuː˩˧] 'women', and 負 [fuː˨] 'negative'), in order to minimise the syllable effect. The target words were embedded in different positions in two carriers phrases: [ŋɔː˩ jiː˨ kaː˥ tʊk˨ __ t͡siː˨] 'I am reading the __ character', and [liː˥ kɔː˧ t͡siː˨ hɐi˨ __ ] 'This character is __'. The 12 sentences (six syllable × two carriers) were repeated ten times, generating a total of 120 trials. Participants were instructed to read the sentences aloud as naturally as they can at a normal speech rate, and the speech outputs were recorded by PRAAT [68] with a Shure SM48 microphone in a soundproof room.

**Tone discrimination.**   The tone discrimination task was also identical to that in [12]. Again, the syllable [fu] carrying six tones was used to control the syllable effect. The six words were recorded in isolation by a female native speaker of Hong Kong Cantonese who does not merge any tones in production and were segmented from the recordings and normalized to 500 ms in duration. Fig 2 shows the F0 of the vocalic portion of the six stimuli at 10 normalized time points. An AX discrimination paradigm was adopted. Two tones were paired with an inter-stimulus-interval of 500 ms. Six AA pairs and 30 AB pairs were repeated ten times to generate a total of 360 trials. Participants were instructed to indicate whether the tones presented were the same or different by pressing the left button (same) or the right button (different) on a computer keyboard within 5 seconds. Both the accuracy and reaction time (RT) were collected. All stimuli were presented on Presentation 20.0 through JVC HA-D610 stereo headphones binaurally in a soundproof room at a comfortable listening level selected by the participant at the beginning and kept constant throughout the experiment.

## Data analysis

For the MBEA test, the scores obtained in each block were first calculated into accuracies, and then analyzed with two-way repeated measures ANOVA, with *group* (amusics, controls, [+per][-pro], [-per][+pro] and [-per][-pro]) as a between-subjects factor, and *subtest* (scale,

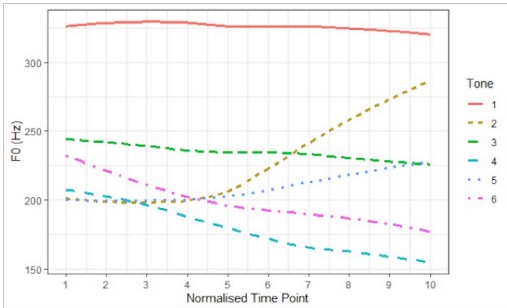

**Fig 2. F0 curve of the six lexical tone stimuli across ten normalized time points.**

contour, interval, rhythm, metre, and memory) as a within-subjects factor. Greenhouse-Geisser method was used to correct the violation of sphericity whenever appropriate.

For the pitch threshold task, two participants from the [+per][-pro] group, one participant from the [-per][+pro] group and one participant from [-per][-pro] group were not included in the analysis due to technical problems in data storage. The data input for analyses were the thresholds in semitone. Three-way repeated measures ANOVA with *group* (amusics, controls, [+per][-pro], [-per][+pro] and [-per][-pro]) as a between-subjects factor and *stimulus type* (speech and nonspeech) and *contour type* (discrete and gliding) as two within-subjects factors was conducted on the pitch thresholds. Greenhouse-Geisser method was used to correct the violation of sphericity where needed.

For the tone production task, two amusics, two controls, one [-per][+pro] participant and three [-per][-pro] participants were not included in the analysis due to technical problems in data storage. The vocalic portion of the target words was manually segmented by the second author and the F0 was measured at 10 time points over the entire course of the vocalic portion in PRAAT. The analyses focused on the production of T2-T5, T4-T6 and the acoustic tone space. Recall that the three tone-merging groups in the current study included [+per][-pro] (T2-T5), [-per][+pro] (T4-T6), and [-per][-pro] (T2-T5). In other words, the only tone pair undergoing tone merging in *production* is T2-T5. For the T4-T6 pair, the [-per][+pro] (T4-T6) group only exhibited perceptual confusion of this pair but not its production. Therefore we did not anticipate the confusion of T4 and T6 in production in any tone-merging group. Nonetheless, acoustic analyses were conducted to verify these impressionistic characterizations, and to further compare the three tone-merging groups to amusics. For T2 and T5, following previous studies [14,69], the following three F0 parameters were calculated: *F0 offset* (F0 at the 10th time point), *F0 slope* (maximal F0 minus minimal F0), and *F0 height* (mean F0 across ten time points). The difference between T2 and T5 (T2 minus T5) were obtained for each of the three aforementioned parameters accordingly to investigate whether there is any merging of T2 and T5 in any group of participants. For T4 and T6, we analysed the *F0 height* (mean F0 across ten time points) and calculated its difference between the two tones (T4 minus T6) to examine if there is any merging of these two tones. In addition, we analyzed the acoustic tone space span of each participant, to examine whether there is a shrinking of the overall tone space. Tone space was obtained by subtracting the lowest F0 of T4 from the highest F0 of T1 in each participant [70]. Any creaky voice or anomalies in the F0 values were excluded from analyses. A total of 133 F0 samples (1.7% of the total F0 samples) were discarded for this reason, which mostly concerned the T4 production (117 samples or 1.5% of the total F0 samples). For all F0 measures, the raw F0 values were normalized using the log-z score method [71], before being input into the analyses. A series of one-way ANOVAs were then conducted with *group* (amusics, controls, [+per][-pro], [-per][+pro] and [-per][-pro]) as a between-subjects factor on the previously mentioned parameters (F0 offset/slope/height differences for T2-T5, F0 height difference for T4-T6, and the tone space) as dependent variables.

For the tone discrimination task, one control participant's data were not included due to technical failure during data storage. The results were first sorted into 21 tone pairs, comprising six same-tone pairs (i.e., T1-T1, T2-T2, T3-T3, T4-T4, T5-T5 and T6-T6) and 15 different-tone ones (i.e., T1-T2, T1-T3, T1-T4, T1-T5, T1-T6, T2-T3. . . T5-T6). The d' scores were used to calculate the participants' sensitivity instead of accuracy, in order to control for response bias [72]. The d' scores were calculated as the z score of the hit rate ('different' responses to different-tone pairs, e.g., T1-T2 and T2-T1) minus that of the false alarm rate ('different' responses to same-tone pairs, e.g., T1-T1 and T2-T2) for each of the 15 different-tone pairs to indicate the perceptual sensitivity. In addition, RT, which has been found to differ between merging and non-merging individuals in previous studies [12,19], was analysed. For the RT

analysis, trials with no responses, incorrect trials and trials with RTs beyond ±3 SD were excluded. Only RTs of different-tone pairs were analyzed. Included trials were then analyzed with two-way repeated measures ANOVA, with *group* (amusics, controls, [+per][-pro], [-per][+pro] and [-per][-pro]) as a between-subjects factor, and *tone pair* (T1-T2, T1-T3 . . . T5-6, a total of 15 different-tone pairs) as a within-subjects factor. Greenhouse-Geisser method was used to correct the violation of sphericity where needed.

Lastly, correlation and regression analyses were conducted to examine the extent to which individual differences in musical abilities and pitch sensitivities can predict the participants' tone discrimination and production performance. In order to reduce the number of predictors and minimize collinearity, a pitch composite score was calculated by averaging the accuracy in the three pitch-based MBEA subtests (scale, contour and interval) for each participant; like-wise, a rhythm composite score was calculated by averaging the accuracy in the two rhythm-based MBEA subtests (rhythm and meter) for each participant. As for the pitch threshold task, since the results showed no difference between the speech and nonspeech stimuli across all participants (see the Results section below), the threshold was averaged across the speech and nonspeech stimuli for the discrete and gliding condition respectively for each participant. Thus, a total of five predictors were included–MBEA-pitch, MBEA-rhythm, MBEA-memory, discrete pitch threshold and gliding pitch threshold. As for the predicted variables, we focused on the overall RT (averaged across all tone pairs) of tone discrimination, which has been found to be explained by non-linguistic working memory and attentional switch abilities in a previous study [12], in addition to the overall d' scores (averaged across all tone pairs). For tone production, we focused on the *F0 offset* difference of T2-T5, which showed the clearest differences among the three parameters (see the Results section below), the *F0 height* difference of T4-T6, and the tone space.

Prior to the regression analysis, bivariate Pearson correlations (two-tailed) were conducted among the five predictors and each of the predicted variables, collapsing the five groups. Only predictors that showed significant correlations with a certain predicted variable were then entered into a linear regression model to determine the relative contribution of each predictor, so as to minimize multicollinearity between the predictors. The predictors were entered into the regression model in a stepwise manner.

## Results

### MBEA

The results of six MBEA subtests of the five groups are shown in Fig 3. A *group × subtest* repeated-measures ANOVA revealed a main effect of *group* ($F(4, 68) = 44.134$, $p < .001$) and *subtest* ($F(2.658, 180.742) = 13.991$, $p < .001$). No significant *group × subtest* interaction was found. Post-hoc pairwise comparisons with Bonferroni correction among the five groups showed that the amusics performed significantly worse than the three tone-merging groups and controls ($ps < .001$), while the three tone-merging groups performed comparably to controls ($ps > .05$). Furthermore, none of the participants in the three tone-merging groups received a global MBEA score below the cut-off score to reach the identification criterion for amusia (the lowest being 77). There is, thus, no evidence that the tone-merging groups had inferior musical abilities compared to the controls. As for the main effect of *subtest*, pairwise comparisons with Bonferroni correction showed that the five groups performed better in the Memory subtest than the other subtests ($ps < .001$), and relatively poorly in the Metre subtest ($ps < .05$, except for Interval). It indicates that the Metre subtest was relatively more difficult and the Memory subtest was the easiest for all participants regardless of groups.

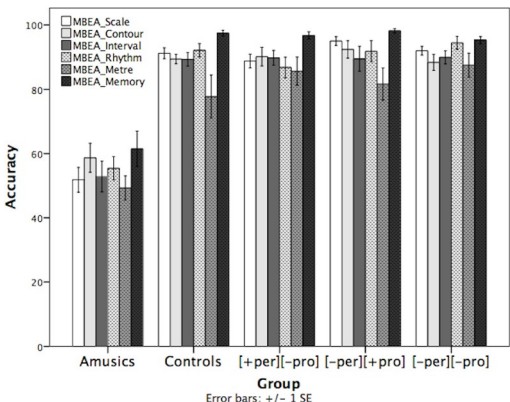

**Fig 3. Performance of the five groups in the six subtests of MBEA (Error bar = ±1 SE).** [+per][-pro]: T2-T5; [-per][+pro]: T4-T6; [-per][-pro]: T2-T5.

## Pitch threshold

The pitch thresholds of the five groups in semitones are shown in Fig 4. A *group × stimulus type × contour type* repeated measures ANOVA found significant main effects of *group* ($F(4, 64) = 7.756, p < .001$) and *contour* ($F(1, 64) = 12.696, p < .001$), but no main effect of *stimulus type*. No interaction effects were significant. Post-hoc pairwise comparisons with Bonferroni correction revealed that the amusics had significantly higher (less sensitive) overall pitch thresholds than the three tone-merging groups and controls ($ps < .002$), while the three tone-merging groups had comparable pitch thresholds to the controls. All five groups had lower (more sensitive) thresholds for the discrete stimuli than the gliding stimuli ($p < .001$).

## Tone production

The normalized F0 trajectories of the six Cantonese lexical tones of the five groups are illustrated in Fig 5. Regarding the production of T2 and T5, a series of one-way ANOVAs with *group* as a factor was conducted on the F0 *offset*, *F0 slope* and *F0 height* difference (T2 minus T5). Effects were reported at the adjusted significance level of $p < .0167$ (.05/3) taking into consideration correction for multiple comparisons. Significant group differences were found

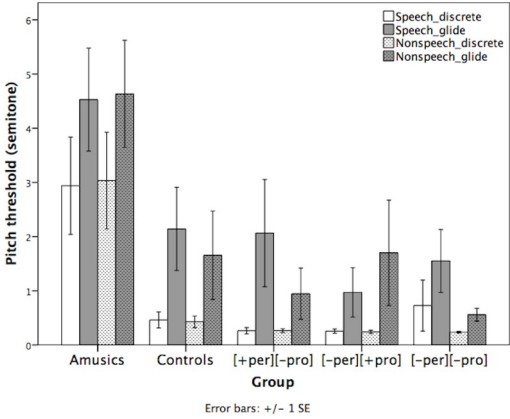

**Fig 4. Pitch thresholds of the five groups in the speech and nonspeech stimuli with discrete and gliding pitch (Error bar = ±1 SE).** [+per][-pro]: T2-T5; [-per][+pro]: T4-T6; [-per][-pro]: T2-T5.

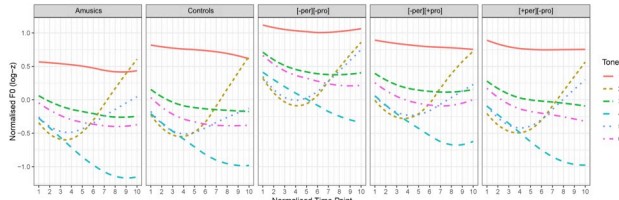

**Fig 5. Normalized F0 trajectories of the six lexical tones produced by the five groups.** [+per][-pro]: T2-T5; [-per][+pro]: T4-T6; [-per][-pro]: T2-T5.

in all three parameters (*F0 offset*: $F(4, 60) = 6.087$, $p < .001$; *F0 slope*: $F(4, 60) = 5.355$, $p < .001$; F0 height: $F(4, 60) = 5.276$, $p = .001$). Post-hoc pairwise comparisons with Bonferroni correction showed that in all three parameters, no significant difference was found between amusics and controls, suggesting that amusics' production of the T2-T5 distinction was comparable to that of the controls, and that no detectable merging of T2-T5 in production was demonstrated in amusics. Furthermore, the [-per][+pro] group (T4-T6) performed comparably to controls in the production of T2-T5 across all three parameters. As expected, the production of T2 and T5 by the [+per][-pro] and [-per][-pro] groups, which concern the merging of T2 and T5 in production only or in both production and perception, displayed reduced differentiation of T2 and T5 as indicated by the significant differences across all three parameters compared with the controls ([+per][-pro] versus controls: F0 offset: $p = .010$; F0 slope: $p = .009$; F0 height: $p = .011$; [-per][-pro] versus controls: F0 offset: $p < .001$; F0 slope: $p = .003$; F0 height: $p = .002$). Furthermore, the [-per][-pro] group, but not the [+per][-pro] group, produced significantly reduced T2-T5 distinction in *F0 offset* compared to amusics ($p = .023$), and the difference between the [-per][-pro] group and amusics was marginally significant for *F0 slope* ($p = .052$). No other effects were significant.

With regard to the production of T4 and T6, one-way ANOVA conducted on the *F0 height* difference of T4 and T6 (T6 minus T4) revealed no significant group difference ($F(4, 60) = .934$, $p = .45$). Following the suggestion of an anonymous reviewer, we also examined the T4-T6 slope difference (T4 minus T6). One-way ANOVA again showed no significant group difference ($F(4, 60) = .630$, $p = .643$). This result confirmed the impressionistic characterization that no merging of T4-T6 in production was demonstrated in any of the three merging groups, amusics or controls.

As for the tone space, the results of one-way ANOVA revealed no significant difference among the five groups ($F(4, 60) = 1.789$, $p = .143$), suggesting that the overall tone space of the three merging groups and amusics remained unaffected despite the diminished F0 contrast of T2-T5 in the [+per][-pro] and [-per][-pro] groups.

## Tone discrimination

The d' scores of the 15 different-tone pairs by the five groups are presented in Fig 6A. A *group × tone pair* ANOVA on the d' scores demonstrated a main effect of *group* ($F(4, 67) = 10.772$, $p < .001$), that of *tone pair* ($F(5.635, 377.539) = 28.907$, $p < .001$), as well as an interaction between *group* and *tone pair* ($F(22.54, 377.539) = 6.064$, $p < .001$). A series of one-way ANOVAs were conducted to examine the group effect within each tone pair. The results revealed that the amusics were less sensitive than the controls in the discrimination of all tone pairs except for T1-T3 (see Table 2 for the full comparison of amusics versus the three tone-merging groups and controls across all tone pairs), and performed comparably inferiorly on T2-T5 and T4-T6 as the [-per][-pro] (T2-T5) and [-per][+pro] (T4-T6) group, respectively. The [-per]

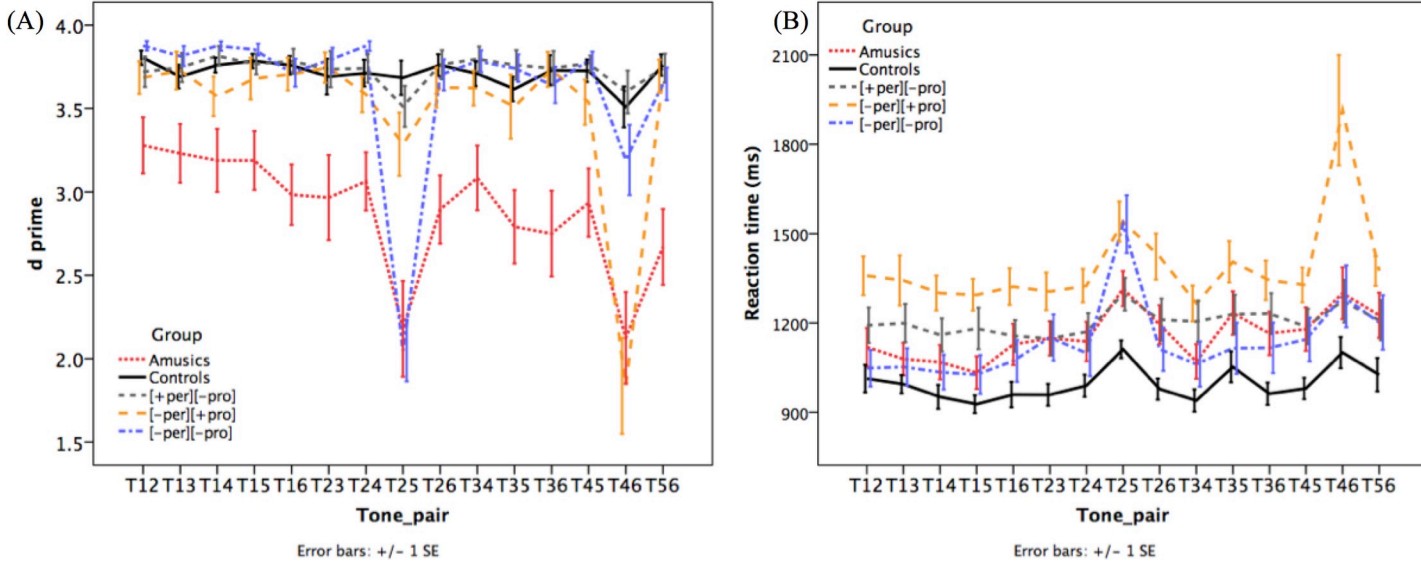

**Fig 6.** The d' scores (A) and reaction time (B) of 15 tone pairs in the tone discrimination task by the five groups (Error bar = ±1 SE). [+per][-pro]: T2-T5; [-per][+pro]: T4-T6; [-per][-pro]: T2-T5.

[+pro] group was less sensitive to only T4-T6 than the controls ($p = .006$), while their sensitivity to other tone pairs were comparable. Similarly, the [-per][-pro] group was less sensitive to only T2-T5 than the controls ($p < .001$), while being comparably sensitive to other tone pairs. On the contrary, the [+per][-pro] group performed comparably to the controls in all tone pairs, with no significant difference in any tone pair. The perceptual confusion patterns reported above match the defining characteristics of each tone-merging group.

**Table 2. Pairwise comparisons of amusics vs. the three tone-merging groups and controls on the d' scores of the 15 tone pairs in the tone discrimination task.**

|  | Amusics vs. [+per][-pro] (T2-T5) | Amusics vs. [-per][+pro] (T4-T6) | Amusics vs. [-per][-pro] (T2-T5) | Amusics vs. Controls |
|---|---|---|---|---|
| T1/T2 | n.s. ($p = .06$) | n.s. | $p = .002$** | $p = .01$* |
| T1/T3 | $p = .034$* | n.s. | $p = .006$** | n.s. |
| T1/T4 | $p = .004$** | n.s. | $p < .001$*** | $p = .010$* |
| T1/T5 | $p = .007$** | n.s. ($p = .053$) | $p < .001$*** | $p = .004$** |
| T1/T6 | $p < .001$*** | $p = .002$** | $p < .001$*** | $p < .001$*** |
| T2/T3 | $p = .016$* | $p = .023$* | $p = .005$** | $p = .023$* |
| T2/T4 | $p = .001$** | $p = .039$* | $p < .001$*** | $p = .002$** |
| T2-T5 | $p < .001$*** | $p = .006$** | n.s. | $p < .001$*** |
| T2/T6 | $p < .001$*** | $p = .007$** | $p < .001$*** | $p < .001$*** |
| T3/T4 | $p = .002$** | n.s. ($p = .06$) | $p = .001$** | $p = .007$** |
| T3/T5 | $p < .001$*** | $p = .024$* | $p < .001$*** | $p = .002$** |
| T3-T6 | $p < .001$*** | $p = .002$** | $p = .002$** | $p < .001$*** |
| T4/T5 | $p < .001$*** | $p = .033$* | $p < .001$*** | $p < .001$*** |
| T4-T6 | $p < .001$*** | n.s. | $p = .006$** | $p < .001$*** |
| T5/T6 | $p < .001$*** | $p < .001$*** | $p < .001$*** | $p < .001$*** |

*** $p < .001$;

** $p < .01$;

* $p < .05$.

Fig 6B displays the RT of 15 different tone pairs by the five groups. A *group* x *tone pair* ANOVA revealed a main effect of *group* ($F(4, 65) = 4.542$, $p = .003$), a main effect of *tone pair* ($F(5.49, 356.868) = 32.989$, $p < .001$) and a *group* x *tone pair* interaction ($F(21.961, 356.868) = 3.817$, $p < .001$). A series of one-way ANOVAs were conducted to investigate the group difference in each tone pair. Significant group effects were detected on every tone pair ($ps < 0.05$) except for T5-T6, where the group difference was marginally significant ($p = 0.054$). The most notable pattern is that the [-per][+pro] (T4-T6) group spent significantly longer time discriminating all tone pairs than the controls, including T4-T6 ($ps < .05$), which they confuse in perception. As a matter of fact, the [-per][+pro] group exhibited longer RT on discriminating T4-T6 than all the three remaining groups ([+per][-pro], [-per][-pro] and amusics) ($ps < .05$). Note that this pattern deviates from the d' scores above, for the reason that the [-per][+pro] group was less sensitive to only T4-T6 than the controls, yet their response speed was slower than controls across all tone pairs. The [-per][-pro] (T2-T5) group was significantly slower than the controls only in the discrimination of T2-T5 ($p < .05$), whereas there was no significant difference between the [+per][-pro] (T2-T5) group and controls in the discrimination of T2-T5. This difference between the [-per][-pro] and [+per][-pro] groups concerning the RT of T2-T5 is aligned with the pattern of d' scores above, and compatible with the defining characteristics of the two tone-merging groups respectively. Furthermore, the [+per][-pro] (T2-T5) group performed similarly to the controls on almost all tone pairs, except for T1-T5 and T3-T4, where they showed significantly longer RT ($ps < 0.05$). Amusics showed a trend of longer RT than controls, but the difference was not significant ($ps > .05$).

## Regression analysis

Prior to the regression analysis, bivariate Pearson correlations (two-tailed) were conducted among the five predictors concerning musical and pitch abilities (MBEA-pitch, MBEA-rhythm, MBEA-memory, discrete pitch threshold and gliding pitch threshold) and each of the predicted variables (the overall d' scores and overall RT of tone discrimination, and the *F0 offset* difference of T2-T5, the *F0 height* difference of T4-T6 and the tone space for tone production), collapsing the five groups. Results of the correlation analyses are shown in Table 3.

For the overall d' scores, all five predictors of musical abilities and pitch sensitivities reached significance ($ps < .01$). No significant correlation was found for the overall RT.

As for tone production, only the MBEA-memory score was significantly correlated with the tone space ($p < .05$), whereas no significant correlation was detected for the F0 offset difference of T2-T5 or the F0 height difference of T4-T6.

Linear regression analyses were then conducted on the overall d' scores of tone discrimination and the tone space for tone production, respectively, with only the significant predictors revealed by the correlation analyses above entered into the model in a stepwise manner.

For the overall d' scores, only the pitch-composite score remained in the model, accounting for 36.2% of the variance in the data (adjusted $R^2 = 0.362$, $p < .001$). The collinearity of the predictors remaining in the model was checked, and the variance inflation factor (VIF) was 1, indicating low multicollinearity [73]. However, it appears that the effect of the pitch-composite score was mainly driven by the amusia group, as the model on the overall d' scores was no longer significant after removing the amusics from the data.

As for the tone space, only the MBEA-memory scores were entered into the model, which significantly accounted for a small portion of variance in the data (adjusted $R^2 = 0.053$, $p = .036$). The VIF (VIF) was 1, indicating low multicollinearity. If excluding amusics from the data, the model remained significant, with MBEA-memory explaining 6.7% of the variance in the data (adjusted $R^2 = 0.067$, $p = .042$). Better MBEA-memory is associated with larger tone

**Table 3. Bivariate Pearson correlations (two-tailed) between the musical and pitch abilities (MBEA-pitch, MBEA-rhythm, MBEA-memory, discrete pitch threshold and gliding pitch threshold) and tone discrimination and production performance (the overall d' scores and overall RT of tone discrimination, and the *F0 offset* difference of T2-T5, the *F0 height* difference of T4-T6 and the tone space for tone production).**

| | | MBEA-pitch composite | MBEA-rhythm composite | MBEA-memory | Discrete pitch threshold | Gliding pitch threshold |
|---|---|---|---|---|---|---|
| Tone discrimination | Overall d' | .570** | .520** | .511** | −.445** | −.413** |
| | Overall RT | .121 | .157 | .180 | .014 | −.079 |
| Tone production | T2-T5 (F0 offset difference) | −.028 | .043 | −.112 | .050 | .111 |
| | T4-T6 (F0 height difference) | −.095 | −.027 | −.069 | .021 | −.033 |
| | Tone space | −.219 | −.229 | −.261* | −.025 | .031 |

* *p* < .05,

** *p* < .01.

space. Note that the effect of MBEA-memory is at most modest, which may require replication by future studies.

## Discussion

In the current study we compared tone merger and amusia by examining the performance of three tone-merging groups who showed various tone perception and production profiles ([+per][-pro] (T2-T5), [-per][+pro] (T4-T6) and [-per][-pro] (T2-T5)) with a group of Cantonese-speaking amusics and another group of controls who did not exhibit tone merging or amusia on a series of musical perception, pitch threshold, tone discrimination and tone production tasks. We discussed the results below in relation to the three research questions raised at the beginning of this paper: (1) whether tone-merging individuals have poor musical perception and reduced domain-general pitch sensitivity (enlarged pitch threshold); (2) whether individual differences in musical abilities and domain-general pitch sensitivities in tone-merging individuals can explain their merging behavior in perception or production; and (3) whether the tone confusion patterns of amusics in tone discrimination and production are comparable to those of tone-merging individuals.

### Musical perception and pitch threshold

The analyses of MBEA scores showed that amusics performed markedly worse than controls and all three tone-merging groups, while the latter were comparable to the controls, and received a global score above the cut-off score to be considered amusics [35,65]. The comparable pitch thresholds of the three tone-merging groups with the controls further support that the ability to perceive just-noticeable differences in pitch height and contour is largely intact in the merging groups. On the other hand, amusics exhibited worse musical perception as well as enlarged pitch thresholds than the other groups. Further results from the regression analyses showed that none of the measures of musical perception or pitch sensitivity predicted the performance of the three merging groups and controls (excluding amusics) in tone discrimination, in terms of their overall d' scores and RT. As for tone production, a small effect of MBEA-memory was found in predicting the tone space (i.e., the F0 span between the highest tone T1 and lowest tone T4) of the three merging groups and controls. This result might suggest that better long-term incidental memory of musical melodies, as assessed by the MBEA-memory subtest, is associated with better production of lexical tones, which also depends on long-term memory of pitch representations. However, this effect is at most modest, and

replication by future studies is needed before any meaningful interpretation can be made. Taken together, these results indicate that tone merging, regardless of their types, is not likely to result from deficits in musical perception or domain-general perceptual acuity to refined differences in pitch height or contour. As such, the primary cause of tone confusion in the tone-merging groups may lie elsewhere other than musical or domain-general pitch-processing deficiencies. We will further return to the discussion of the underlying cause of tone merger and its relationship with amusia, after discussing the performance of amusics in tone production and discrimination in the next subsection.

It is worth noting that amusics demonstrated enlarged (less sensitive) pitch thresholds than controls in all four conditions (speech/nonspeech × discrete/gliding). This observation partly deviated from the patterns observed in Mandarin-speaking amusics, who were found to exhibit higher thresholds for discrete tones and comparable thresholds for contour tones to controls [39,74]. Besides, all five groups of Hong Kong Cantonese speakers demonstrated larger thresholds for detecting pitch patterns for contour tones than for discrete tones, another pattern differing from Mandarin speakers [39]. These discrepancies may be explained by the difference in their tonal inventories. Whereas there are multiple level tones in Hong Kong Cantonese that primarily contrast in F0 height, the four tones in Mandarin can be largely distinguished by F0 contour [75]. As a result, Cantonese speakers are presumably more *perceptually* sensitive to small pitch height differences, whereas Mandarin speakers demonstrated better *perceptual* sensitivity to pitch contour differences (in terms of tone production Mandarin speakers have been found to demonstrate sensitivity to F0 height, showing F0 consistency in tone production [54]). This explanation is also in line with the findings of several cross-linguistic studies on tone perception that Cantonese listeners placed more weight on pitch height cues, in contrast to Mandarin listeners who placed more weight on pitch direction cues [76–78]. Furthermore, an event-related potentials (ERPs) study revealed delayed mismatch negativity (MMN) and reduced MMN amplitude toward a native tone pair with a pitch contour contrast (T1-T2, high level-high rising tone) compared to another tone pair with a pitch height contrast (T6-T1, low level-high level tone) in Cantonese listeners [79], which also supports the notion of greater sensitivity of Cantonese listeners to pitch height differences. Together with the findings from Mandarin speakers, these results further imply that listeners' psychophysical pitch thresholds are malleable and may be subject to the influence of their native tonal language backgrounds.

## Lexical tone production and discrimination

The analyses of lexical tone production and discrimination revealed further differences between amusics and the three tone-merging groups. Regarding tone production, no significant difference was found between amusics and controls in the T2-T5 distinction (as measured by the *F0 offset*, *F0 slope* and *F0 height* difference), the T4-T6 distinction (as measured by the *F0 height* difference), or the tone space. This result suggests that tone production is largely intact in amusics, which replicates the previous study on Cantonese speakers with amusia [26], providing converging evidence for preserved tone production in amusics [35,50]. As for the tone-merging groups, the [+per][-pro] (T2-T5) and [-per][-pro] (T2-T5) groups produced reduced T2-T5 distinction (in terms of *F0-offset*, *F0-slope* and *F0-height* differences) compared to the controls, a result parallel with previous studies [14].

As for tone discrimination, the d' scores showed that the confusion patterns exhibited by the [-per][+pro] (T4-T6) and [-per][-pro] (T2-T5) groups were highly selective to T4-T6 and T2-T5, respectively. These selective patterns are somewhat expected, but interesting nonetheless, because tone-merging participants were identified based on inclusionary criteria (i.e., the

participants were assigned into the tone-merging groups as long as they confused the concerned pair of tones in production, perception or both), but not on exclusionary criteria (i.e., the participants would not be excluded if they confused other pairs of tones). With regard to amusics, it is worth noting that they performed not only equally poorly in discriminating T4-T6 and T2-T5 as the two merging groups, but also in other tone pairs with large acoustic differences such as T1/T6 (high-level vs. low-level) and T2/T4 (high-rising vs. extra-low-level/ low-falling), demonstrating low sensitivity to tone distinctions across the board compared to the controls. This result is compatible with the finding of overall lower accuracy in lexical tone discrimination among Cantonese-speaking amusics in previous studies [26,43]. Altogether, these results clearly demonstrate different profiles of lexical tone production and discrimination in amusics and tone-merging individuals, with broadly impaired tone discrimination and largely preserved tone production in amusics and highly selective confusion in tone perception and/or production in tone-merging individuals.

In addition, the analyses of the discrimination RT revealed two interesting patterns that have not previously been reported and demanded an explanation. First, while amusics performed comparably inferiorly on the d' scores of discriminating T4-T6 and T2-T5 as the [-per] [+pro] (T4-T6) and [-per][-pro] (T2-T5) groups compared to the controls, the RT of amusics was not significantly longer than that of controls. On the other hand, these two merging groups exhibited reduced sensitivity as well as slower RT at discriminating T2-T5 and T4-T6, respectively, compared to the controls. Thus, there appears to be some misalignment between the discrimination accuracy (in terms of d' scores) and response speed, and possibly a speed-accuracy tradeoff in the performance of amusics. Note that this pattern deviated from the finding of concomitant impairments in (lower) discrimination accuracy and (longer) RT in amusics in previous studies [43,46]. One possible explanation is that the amusic participants recruited in the current study (MBEA global score of amusics: $M = 54.9$, $SD = 16.3$; also see Fig 3) have rather severe musical perception impairments compared to those amusics recruited in previous studies [43,46] (Global score of amusics: $M = 62.4–65.4$, $SD = 3.9–8.6$). Although typically listeners would spend longer time in a challenging task, for the amusic participants who have a profound pitch-processing deficit, it is possible that the tone discrimination task was so perceptually challenging that they did not spend as much time on it. It should be noted that the two earlier studies [43,46] used the Online Identification Test of Congenital Amusia to identify amusics, which is a shorter musical assessment test compared to the MBEA. This speculation should be further investigated in future studies that examine the effect of severity of musical perception deficiency in amusics on lexical tone discrimination using the same musical assessment test.

A second pattern that is worth noting is that the [-per][+pro] (T4-T6) participants exhibited the longest RT among all five groups. The 'near merger' phenomenon (i.e., poor perception but good production) first proposed by Labov and colleagues [80] has been baffling since it challenges the traditional models for phonological processing, where good production presumes the existence of relevant sensory representations associated with good speech perception. Law and colleagues [16] proposed that the degradation of sensitivity in perceiving the T4-T6 contrast in the [-per][+pro] group is a consequence of top-down processing in language communication. Listeners may rely on contextual information in the recognition of lexical items during speech communication, and thus acoustic input may not undergo complete analysis, which in turn weakens their sensitivity to speech sound contrasts, in particular with tone pairs of small acoustic differences such as T4-T6 and T2-T5. In a tone discrimination task like in the current study, where contextual information is absent and bottom-up acoustic analysis is crucial, it is likely that the [-per][+pro] participants, who have a tendency to rely on top-down processing, did not fare well, thereby exhibiting overall longer RTs in tone discrimination.

One last point to note is the production-perception dissociation in amusics. The results revealed that amusics' lexical tone production was largely intact despite their poor performance in lexical tone and musical perception. Such dissociation in tone perception/production in amusics has been repeatedly reported in previous studies. Yang and colleagues [50] found that tone production in Mandarin-speaking amusics was spared although their musical tone perception was poor. Cantonese-speaking amusics in [26] showed normal production of native tones with subtle pitch height and contour differences amid inferior tone perception. Even Mandarin tone agnostics, who exhibited exceptionally poor tone perception ability, were found to be intact in tone production [35]. It has been proposed that the production-perception de-coupling in amusics can be explained by reduced arcuate fasciculus connectivity that connects the neural circuitries of sound production and perception, and a distinct and preserved auditory action stream in the amusical brain [81–83]. The discussion here also provokes interesting but currently unexplored questions regarding the neural bases for the production-perception dissociation in individuals with near merger who exhibit highly restricted dissociation for specific tonal contrasts. Future neuroimaging studies should undertake this investigation and compare in what ways the neural substrates for the production-perception dissociation in individuals with near merger overlap or differ from those of amusics, in order to shed more light on the relationship of tone merger and amusia, and the relationship of speech production and perception.

## Relationship between amusia and tone merger

The comparison of amusics and tone-merging individuals on musical perception, pitch threshold, tone discrimination and tone production tasks above provided converging evidence for the dissociation between tone merger and amusia. These results clearly demonstrate that the auditory pitch deficits of amusics are domain general, which are manifested in musical perception and pitch sensitivity as well as in lexical tone discrimination, and that there is production-perception dissociation in amusics. In contrast, the tone confusion of tone-merging individuals appears to be linguistically based, with their musical perception and fine-grained pitch sensitivities intact, and their tone confusion patterns highly selective in perception and/or production. These findings indicate that tone merger and amusia are inherently different from each other.

In addition to clear behavioral evidence for the dissociation of tone merger and amusia described above, previous studies have provided indirect evidence that the neural bases underlying the perceptual confusion of tones are likely to differ between tone-merging individuals and amusics. Several electrophysiological studies have consistently revealed that conscious detection of pitch changes in active listening conditions is particularly impaired in the amusical brain (e.g., as indexed by the P300), whereas preattentive processing of the same pitch changes in passive listening conditions is largely intact (e.g., as indexed by the MMN) [46,84–87]. In contrast, available evidence indicates that the MMN is diminished in a group of tone-merging individuals ([-per][+pro] (T4-T6)) during passive processing of T4-T6 changes [16]. Thus, it is possible that amusics and tone-merging individuals are differentially affected in conscious and preattentive processing of tone differences, a question that awaits future electrophysiological studies that directly compare amusics and tone-merging individuals within the same design.

How to reconcile the results of the current study with earlier studies on tone merger? Intriguingly, previous studies have found that tone-merging individuals have inferior domain-general attentional switching abilities, which can explain their tone perception performance (e.g., reaction time in tone discrimination) and its neural measure (e.g., MMN latency)

[12,24]. The authors argued that sluggish attentional switch may prevent the automatic attentional system from disengaging itself fast enough from one sound to the next, leading to degraded sensory analysis and auditory representation. Intriguingly, the current study showed that musical perception and domain-general pitch sensitivity are more or less intact in tone-merging individuals. This result complements and corroborates with the earlier finding that musical training had no effect on the occurrence of tone merger, which points to the presence of tone merger independent of superb musical abilities [19]. Altogether, these results reinforce the view that tone merger is not driven by listeners' reduced perceptual acuity to pitch. On the other hand, tone merger may be deemed as an emerging sub-variety of standard Hong Kong Cantonese [13,19]. According to this view, these tonal variations among merging individuals may be analogous to sub-varieties or sociolinguistic variations (e.g., a sociolect) of standard Cantonese. Even among speakers from the same geographical location (e.g., a town), there could be phonetic or lexical differences among individuals graded by socioeconomic classes, age groups or other social groups [88]. In view of the large amount of individual variations concerning the tone pairs undergoing merger and the complex merging patterns in production or perception or both [14], it is likely that these sociolinguistic variations of standard Cantonese are not yet fully established (i.e., mergers in progress [14]). These variations may be partly driven by the highly diverse and multilingual environment in Hong Kong [19], and to some extent the lack of a romanization system of spoken Cantonese in Hong Kong, which has been associated with lower phonological awareness, including tone awareness, in native language processing [89–91]. Future studies may investigate how these factors interact with individual differences (e.g., in attentional switching) in driving the tone variations.

As discussed above, the findings from both studies on musicianship and musical disorder have converged to show that tone merger appears to be more or less independent of the influence of music. Nonetheless, this finding should not be taken as evidence against the cross-domain transfer of speech and music, since there are clear and widely attested effects of musicianship and musical disorder on speech processing, including those on Cantonese [26,27]. Indeed, the current study also replicated the previous finding that individuals with amusia have impaired tone discrimination, supporting the music-to-speech transfer. It is likely that the cross-domain transfer between speech and music is not a simple all-or-none effect, and demonstrating where and how the transfer between speech and music is restricted is equally important for understanding the relationship of speech and music and their neural infrastructure.

## Conclusion

The current study provides the first set of evidence that tone merger and amusia are dissociated with presumably different underlying causes, and that the similarities in tone confusion between them are only superficial. While the auditory pitch deficits in amusia are domain general with impaired tone discrimination across the board but intact speech production, the tone confusion among tone-merging individuals appears to be linguistically based, with highly selective tone confusion patterns that can occur in perception, production or both. While we did not find evidence for a musical perception disorder in the current sample of tone-merging individuals, the possibility cannot be completely ruled out that amusics, who constitute up to 4% of the population, might make up at least a small portion of individuals with perceptual tone mergers (i.e., [+per][-pro] and [-per][+pro]). Future studies with a larger sample size of tone-merging individuals should be carried out to investigate this question. Lastly, it is believed that studies examining the neural bases of tone merger and amusia can further clarify their differences and underlying mechanisms.

## Supporting information

**S1 File.**

(XLSX)

## Author Contributions

**Conceptualization:** Caicai Zhang.

**Data curation:** Oi-Yee Ho, Jinghua Ou.

**Formal analysis:** Caicai Zhang, Oi-Yee Ho.

**Funding acquisition:** Caicai Zhang.

**Investigation:** Caicai Zhang, Oi-Yee Ho, Sam-Po Law.

**Methodology:** Jing Shao, Jinghua Ou.

**Writing – original draft:** Caicai Zhang, Oi-Yee Ho.

**Writing – review & editing:** Jinghua Ou, Sam-Po Law.

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
