## [Decision Letter · Decision Letter 0]

10 Mar 2021

PONE-D-20-39863

Dissociation of tone merger and congenital amusia in Hong Kong Cantonese

PLOS ONE

Dear Dr. Zhang,

Thank you for submitting your manuscript to PLOS ONE. After careful consideration, we feel that it has merit but does not fully meet PLOS ONE’s publication criteria as it currently stands. Therefore, we invite you to submit a revised version of the manuscript that addresses the points raised during the review process.

We look forward to receiving your revised manuscript.

Kind regards,

Psyche Loui

Academic Editor

PLOS ONE

Journal Requirements:

2.We note that you have indicated that data from this study are available upon request. PLOS only allows data to be available upon request if there are legal or ethical restrictions on sharing data publicly. For more information on unacceptable data access restrictions, please see http://journals.plos.org/plosone/s/data-availability#loc-unacceptable-data-access-restrictions.

Reviewers' comments:

Reviewer's Responses to Questions

**Comments to the Author**

1. Is the manuscript technically sound, and do the data support the conclusions?

Reviewer #1: Yes

Reviewer #2: Partly

2. Has the statistical analysis been performed appropriately and rigorously? 

Reviewer #1: Yes

Reviewer #2: Yes

3. Have the authors made all data underlying the findings in their manuscript fully available?

Reviewer #1: Yes

Reviewer #2: No

4. Is the manuscript presented in an intelligible fashion and written in standard English?

Reviewer #1: Yes

Reviewer #2: Yes

5. Review Comments to the Author

Reviewer #1: This paper presents a large and carefully-conducted study on tone merger in Hong Kong Cantonese. In general I recommend publication, but have some comments about the interpretation, and discussion of the related literature.

p. 27 lines 9 – 16. The claim that speakers of Mandarin are less sensitive to tone height is at variance with the findings of Deutsch Henthorn and Dolson, 2004 (Music Perception 2004, 21, 339-356) that speakers of Mandarin showed remarkable pitch consistency in reciting the same list of twelve words on two different days. Out of 15 subjects, five showed averaged pitch differences of less the 0.25 semitone, and another 5 showed averaged pitch differences of 0.25-0.50 semitone. As in Cantonese, both pitch height and contour are determinants of tone in Mandarin.

p.30. lines 2-3, and line 11. The invocation of attention-switching requires explanation. In what way does attention-switching explain the findings? An increase in tone merging among young adult speakers of Cantonese would appear to be a good explanation of the findings. Since several different tone languages are spoken in Hong Kong, one would expect to find an impact of exposure to these different languages. The authors might want to examine this in a further study, in which they investigate the linguistic backgrounds of the subjects in detail.

Reviewer #2: This study investigated the relationship between music and speech by comparing Cantonese amusic participants with Cantonese participants who were merging tones in perception and/or production in a series of tasks testing their musical and lexical tone abilities. They would like to address whether amusia and tone merge were based on domain-general mechanisms or whether they were domain-specific. They found that amusic and merging participants had different patterns which support the domain-specific view. They concluded that their findings corroborate an earlier study which they have discussed in detail in the Introduction to motivate their study.

The study provides novel data which help explore the relationship between music and speech from a less-understood perspective of whether merging participants would have poorer musical abilities. The manuscript is clearly written and easy to follow. There are, however, some issues, mainly methodological, that need to be addressed in the revision, although stronger justification for their study is also needed.

p.6: ‘First, the merging and non-merging participants were selected based on the confusion of

tones in production, and it is not clear whether they showed any perceptual confusion of tones.’ The authors can add ‘given that there are different types of merging speakers as discussed above’.

p.6: ‘Thirdly and most importantly, it has not been investigated before whether merging individuals, especially those who confuse tones perceptually, would show inferior musical abilities and lower sensitivities to fine-grained pitch height and contour differences compared to non-merging individuals.’

This seems to be an important motivation of the study, but there is no clear exposition why this would be expected, if previous studies have shown the separation between music and speech. In addition to Mok & Zuo (2012) who reported merging speakers with advanced musical training, there are other similar studies which the authors did not discuss. For example: Maggu, Wong, Antoniou, Bones, Liu & Wong (2018) Effects of combination of linguistic and musical pitch experience on subcortical pitch encoding. Journal of Neurolinguistics, 47: 145, and the relevant studies reviewed in there. The authors need to elaborate more on why they would have that hypothesis in the first place, even though they have more discussion later about amusia which is not the same issue.

p.10-11: ‘a group with poor perception and good production of T4-T6 or NM, a group with good perception and poor production of T2-T5 or PM, and a group with poor perception and poor production of T2-T5 or FM’ Why did they mix the two merging tone pairs here? Why did they not control for it? Please explain. Also, how about the other merging pair of T3/T6? They did not mention anything about this pair in the study. Why did they exclude it?

p.13: ‘The participants’ tone discrimination accuracy was measured from the AX discrimination task.’ Please give more detail about this AX discrimination task.

p.13: Please explain clearly why 80% was chosen as the threshold for dividing participants into different groups. 80% is well above the chance level (1/6 = 16.67% for all six tones, 1/3 = 33.33% for confusion between the two target tones). It is obviously that those with accuracy lower than 80% had poorer performance than control participants, but whether they should be classified as merging participants need more justification. Why did the authors not use their perception and production accuracy rates as continuous variables? More explanation is needed, as it is important in interpreting their results.

‘Participants who had an accuracy below 80% in T2-T5 discrimination but more than 80% accuracy in T2-T5 production were classified into the PM group (T2-T5 [+per][-pro])’ Shouldn’t this group be classified as [-per][+pro]??? Have I missed something here?

‘those who achieved lower than 80% accuracies in both T2-T5 perception and T2-T5 production were assigned into the FM group (T2-T5 [-per][-pro])’ Classifying someone who can distinguish T2 and T5 well above chance level as the FULL MERGER group is baffling. The same also applies to the PM and NM classifications. I think the authors should not simply adopt the terms used in Fung & Lee (2019) whose claims about the tone pairs being collapsed were too exaggerated.

p.13: ‘Note that tone perception or production confusion is not part of the diagnostic criteria of amusia.’ So were any of their amusic participants merging tones or were any of the merging participants amusic? Please clarify.

p.14-15: The authors can include a figure to illustrate the various patterns in their experiment for easy reference.

p.16: Was this AX tone discrimination task the same as the screening AX task mentioned on p.13??? If not, what’s the difference between the two tasks? Please clarify.

p.18: ‘We did not anticipate the confusion of T4 and T6 in production in any group, even for the NM group whose tone confusion concerns discrimination not production.’ Please clarify why.

p.18: ‘For T4 and T6, we analysed the F0 height (mean F0 across ten time points)’ The difference between T4 and T6 is mainly manifested in the second half of the tone. Why did the authors not include a more dynamic measure (e.g. slope) to compare these two tones but relied on overall F0 height? Please explain.

Also, the second half of T4 is often creaky. How was it treated in the acoustic measurement? This has a bearing on their tone space measurement (Tone space was obtained by subtracting the lowest F0 of T4 from the highest F0 of T1 in each participant) as the lowest F0 of T4 can vary between participants depending on how the creaky parts were treated.

p.29: ‘One possible explanation is that the amusic participants recruited in the current study have rather severe impairments in musical pitch perception (see Figure 2).’ Reference is needed to support that the amusic participants in this study had severe impairments, or with more severe impairment that those in Shao et al., 2016 and Zhang & Shao, 2018.

p.29: ‘A second pattern that is worth noting is that’ They should start a new paragraph here.

6. PLOS authors have the option to publish the peer review history of their article (what does this mean?). If published, this will include your full peer review and any attached files.

Reviewer #1: **Yes: **Diana Deutsch

Reviewer #2: No

---

## [Author Response · Author response to Decision Letter 0]

14 Apr 2021

Please refer to the Response to Reviewers.

---

## [Decision Letter · Decision Letter 1]

25 May 2021

PONE-D-20-39863R1

Dissociation of tone merger and congenital amusia in Hong Kong Cantonese

PLOS ONE

Dear Dr. Zhang,

Thank you for submitting your manuscript to PLOS ONE. After careful consideration, we feel that it has merit and may be published with some additional minor revisions as suggested by the reviewer as shown below. Therefore, we invite you to submit a revised version of the manuscript that addresses the points raised during the review process.

We look forward to receiving your revised manuscript.

Kind regards,

Psyche Loui

Academic Editor

PLOS ONE

Journal Requirements:

Reviewers' comments:

Reviewer's Responses to Questions

**Comments to the Author**

1. If the authors have adequately addressed your comments raised in a previous round of review and you feel that this manuscript is now acceptable for publication, you may indicate that here to bypass the “Comments to the Author” section, enter your conflict of interest statement in the “Confidential to Editor” section, and submit your "Accept" recommendation.

Reviewer #2: All comments have been addressed

2. Is the manuscript technically sound, and do the data support the conclusions?

Reviewer #2: Yes

3. Has the statistical analysis been performed appropriately and rigorously? 

Reviewer #2: Yes

4. Have the authors made all data underlying the findings in their manuscript fully available?

Reviewer #2: Yes

5. Is the manuscript presented in an intelligible fashion and written in standard English?

Reviewer #2: Yes

6. Review Comments to the Author

Reviewer #2: I think the authors have satisfactorily addressed my previous comments. The current manuscript is improved in various ways. Just some final minor comments:

p.4: ‘Overall, three types of tone merger were reported: [+per][-pro] indicating good perception but poor production, [-per][+pro] indicating poor perception but good production, and [-per][-pro] where both perception and production are poor’ It is commendable that the authors have changed the terms they use for the three groups. I think ‘better’ and ‘poorer’ may be more appropriate than ‘good’ and ‘bad’.

p.18: ‘Any creaky voice or anomalies in the F0 values were excluded from analyses.’ They should report how much data was excluded. The lowest F0 in T4 is closely related to the creaky portion. More details should be given here.

p.28: (J. Gandour, 1983; J. T. Gandour & Harshman, 1978; Z. Qin & Jongman, 2016)

p.31: (P. Loui, Alsop, & Schlaug, 2009; Psyche Loui, Guenther, Mathys, & Schlaug, 2008; Victoria J. Williamson, Liu, Peryer, Grierson, & Stewart, 2012)

They should remove the initials. Please check for citation consistence throughout the manuscript.

p.36: ‘One possible explanation is that the amusic participants recruited in the current study have rather severe impairments in musical pitch perception (see Figure 3) (Shao et al., 2016; Zhang & Shao, 2018).’ I think they need to point out explicitly how worse their participants were compared to those in the two previous studies to strengthen their argument.

7. PLOS authors have the option to publish the peer review history of their article (what does this mean?). If published, this will include your full peer review and any attached files.

Reviewer #2: No

---

## [Author Response · Author response to Decision Letter 1]

3 Jun 2021

Please refer to the response to reviewers.

---

## [Editor Report · Decision Letter 2]

17 Jun 2021

Dissociation of tone merger and congenital amusia in Hong Kong Cantonese

PONE-D-20-39863R2

Dear Dr. Zhang,

We’re pleased to inform you that your manuscript has been judged scientifically suitable for publication and will be formally accepted for publication once it meets all outstanding technical requirements.

Kind regards,

Psyche Loui

Academic Editor

PLOS ONE
---

## [Editor Report · Acceptance letter]

21 Jun 2021

PONE-D-20-39863R2 

Dissociation of tone merger and congenital amusia in Hong Kong Cantonese 

Dear Dr. Zhang:

I'm pleased to inform you that your manuscript has been deemed suitable for publication in PLOS ONE. Congratulations! Your manuscript is now with our production department. 

Kind regards, 

on behalf of

Dr. Psyche Loui 

Academic Editor

PLOS ONE